# Polarization-Dependent Scattering of Nanogratings in Femtosecond Laser Photowritten Waveguides in Fused Silica

**DOI:** 10.3390/ma15165698

**Published:** 2022-08-18

**Authors:** Guanghua Cheng, Ling Lin, Konstantin Mishchik, Razvan Stoian

**Affiliations:** 1School of Artificial Intelligence, Optics and Electronics(iOPEN), Northwestern Polytechnical University, Xi’an 710072, China; 2State Key Laboratory of Transient Optics and Photonics, Xi’an Institute of Optics and Precision Mechanics, CAS, Xi’an 710119, China; 3Laboratoire Hubert Curien, UMR 5516 CNRS, Université Jean Monnet, 42000 Saint Etienne, France

**Keywords:** polarization-selective devices, waveguides, femtosecond phenomena

## Abstract

The properties of polarization-selective, light-guiding systems upon subwavelength nanogratings formation in the case of type II refractive index traces induced by femtosecond laser pulses in bulk fused silica were studied. Polarization-dependent scattering is analyzed both in simulation using a finite-difference, time-domain method and in experiments. We argue that the polarization-sensitive optical guiding of type II waveguides is due to polarization-dependent scattering of nanogratings. Optical designs can then be suggested where the guiding efficiency of type I traces can be combined with type II anisotropies. A low-loss waveguide polarizer is demonstrated based on the modulation of the evanescent field emerging from type I waveguides using polarization-dependent scattering of neighboring nanogratings.

## 1. Introduction

Femtosecond (fs) infrared laser pulses focused inside bulk fused silica can deposit energy in extremely localized regions through nonlinear photoionization mechanisms and carrier relaxation, affecting the microscopic glass matrix. The light–matter interaction gives rise to a permanent structural modification associated with a refractive index change [1,2]. This local change of optical properties constitutes the base of laser three-dimensional (3D) optical functionalization. Typically, the modifications in fused silica can be divided into two categories: type I waveguide and type II waveguide. The type I waveguide is written at low pulse radiation levels; the modification is quasi-isotropic, leading to a smooth refractive index increase of about 10^−3^, its losses typically lower than 0.2 dB/cm, which can be useful for photonic device fabrication. The type II waveguide is written at higher pulse energy levels; the laser-induced modification may contain an arrangement of periodic nanoplanes which align themselves orthogonally to the laser polarization, which can be utilized to form type II waveguides [3] with polarization-dependent guiding properties. A typical type II waveguide is shown in Figure 1a, with its near-field modes shown in Figure 1b,c and a scanning electrical microscopy (SEM) image of a cross section shown in Figure 1d.

The transition from type I to type II waveguides is determined by the deposited power, accumulation dose, and pulse duration. The nanoplane modulation region is composed of a periodic assembly of thin layers with a lower refractive index as compared to the pristine material, separated by thick layers of higher refractive index. Typical local refractive index values within the thin and the thick layers are approximately 1.1 and 1.45, respectively, whereas their corresponding thicknesses are about 10 nm and 240 nm. The layers are oriented perpendicular to the laser electric field, with a period of approximately λ/2n (n is the refractive index and λ is the writing wavelength). Such an assembly of nanogratings shows form birefringence [4] and behaves as a negative uniaxial crystal with the optical axis normal to the nanoplanes. The ability to generate periodic nanostructures in the bulk silica has led to the development of a new set of optical functionalities related to polarization guiding, phase retardation, multiscale information storage, and polarization-sensitive devices, ranging from diffraction gratings to radial polarization and optical vortex converters [5,6,7].

In addition to the birefringent properties, the nanoplane structure shows strong polarization-dependent (PD) scattering effects, manifested, for example, in off-axis third harmonic diffraction and anisotropic reflection patterns [3,8]. Several possible applications were already proposed that include polarization-sensitive waveguides, polarization-sensitive routers [3], and 3D marking [9,10]. In polarization-dependent guiding applications involving type II structures, the associated polarization properties were evaluated as a function of the fill factor of the nanostructured domains [11], thus showing an opportunity for optimizing the optical performances. Optimization schemes for laser-induced nanoscale patterning were conducted with programmable time envelopes of the laser excitation pulses by monitoring the diffraction intensity of the nanoplanes in real time [12]. Beside the form birefringence and polarization-sensitive scattering, diffraction of the nanograting has been explored recently. Diffraction gratings at several centimeters on fused silica and silicon wafer with Ag film have been reported [13,14]. In most cases, nanogratings induced by fs laser pulses have been observed in fused silica and some derivates, rarely in other optical glasses. Considering a potential choice for fabricating polarizing elements in integrated photonic circuits, other optical materials such as Ti:sapphire [15], Yttrium Aluminum Garnet (YAG) [16], and Diamond-like carbon [17] enable the formation of the nanograting irradiated by a polarized ultrashort pulse laser.

In view of its potential application, the underlying physical mechanisms of PD scattering require further quantification. In a classical theoretical frame, Rayleigh scattering or elastic scattering of light by particles much smaller than the light wavelength should be a dominant factor. However, it is interesting to investigate the polarization behavior of the particular geometry of dielectric nanoplanes. In this paper, we present a systemic investigation of the fundamental mechanism, the performance, and the applicability of laser-triggered nanogratings. In order to explain the underlining mechanism of PD scattering, finite-difference time-domain (FDTD) simulation was conducted. The simulation results confirm that the scattering cross section of the nanopatterned trace is highly sensitive to the injected polarization. PD scattering was further analyzed in experiments, indicating that scattering is the major source of polarization sensitivity behind the observed waveguiding behavior of type II waveguides, whereas form birefringence is a relatively minor contributor. Coupling mixed regimes of type I and type II, a low-loss waveguide polarizer is demonstrated based on the modulation of the evanescent field near a type I waveguide through polarization-dependent scattering of nanogratings.

## 2. The Mechanism of PD Guiding

In order to understand the PD guiding of a type II waveguide, the mode was analyzed with the finite element analysis method (FEM) first. Subsequently, the PD scattering effect of nanogratings was analyzed with both FDTD simulation and experiments.

### 2.1. Mode Analysis

Based on the data reported in [3,10] and our SEM analysis (Figure 1d), it is reasonable to construct an ideal model of the cross section of a type II waveguide as shown in Figure 2a. The waveguide core in the center has a higher refractive index than the cladding. Nanogratings are written outside the core, shown as the shaded region in the figure. Outside the nanograting region is the unaffected fused silica substrate. According to the data obtained in [3,10], the following parameters were used in the model: the refractive index of cladding is 1.45 at 800 nm (index of fused silica), and there is a 0.1% index step between the core and the cladding. The 10nm thin layer of the nanograting has the refractive index of 1.1, whereas the thick layer is 240nm with the refractive index of 1.45. The radii of the waveguide core and the nanograting region are 1.5 μm and 3.5 μm, respectively.

The modes of type II waveguide were computed using the finite element analysis method. The FEM simulations were performed using the COMSOL simulation package. Figure 2b shows the flux profile, (the z-component of the Poynting vector) of the fundamental x-polarized mode, whereas Figure 2c shows the energy flux profile of the fundamental y-polarized mode. It is clear that the structure can support the fundamental mode in both polarizations. The effective indices calculated for the x and y-polarized modes were n_eff. x_ = 1.44478 and n_eff. y_ = 1.44371, respectively, which demonstrates a mode birefringence with a magnitude of Δn_eff_ = n_eff. x_ − n_eff. y_ = 0.00107. Here, the mode birefringence is primarily attributed to the planar nanogratings in the horizontal direction, which break the circular symmetry of the waveguide. However, as mentioned in the introduction, a nanoscale modulated structure shows form birefringence if the layer period is smaller than the wavelength of light in the medium. Thus, if we regard the nanograting as an equivalent crystal, mode birefringence may also be caused by the material anisotropy of the waveguide.

### 2.2. FDTD Simulation

It is concluded from the mode analysis that type II waveguides should have two orthogonally polarized fundamental modes. The form birefringence of the nanogratings contributes to the mode birefringence of the two modes. As in a fiber polarizer [18,19,20], differential propagation losses exist between these two modes, which effectively leads to the PD guiding. In the present case, the loss difference is caused by the PD scattering of the nanogratings. The mechanism and the impact of PD scattering that affect the PD guiding of type II waveguide are discussed below.

Practical nanogratings can be viewed as being composed of many structured nanoplanes oriented perpendicular to the writing polarization embedded in a uniform dielectric material. The scattering of a single nanoplane subject to the incident polarized light can be numerically simulated by the FDTD method. The shape of a single nanoplane is best described as an oblate cuboid. The three-dimensional FDTD simulations were performed using a commercial FDTD simulator (FDTD Solutions version 6.5) [21]. In the simulation, a plane wave total-field scattered-field (TFSF) source is used. The surfaces of the box in the middle are TFSF boundaries inside which the total fields are available and outside which only the scattered fields are available, as shown in Figure 3. The pink arrow in the figure shows the direction of incident wave propagation (z) and the blue arrow shows the orientation of polarization. The oblate cuboid (10 × 100 × 100 nm) is in the center of the simulation region. Six power monitors are located on the surfaces of the largest box to measure the scattered field, and six additional power monitors are located on the surfaces of the smallest box to measure the total field. The scattering cross-section is defined as δ_scat_(ω) = P_scat_(ω)/I_source_(ω), where P_scat_ is the total scattered power in [W] and I_source_ is the incident intensity in [W/m^2^]. In the simulation, the refractive index of the cuboid is 1.1, and it is assumed to be dispersionless between 750 nm and 850 nm. The background refractive index of 1.45 is used in the simulation. For such a single nanoplane, the scattering intensity in the far-field zone in the x-y plane for both excitation polarizations are shown in Figure 3. The information is complemented in Figure 4, which summarizes the quantitative scattering cross sections in the two orthogonal directions, perpendicular or parallel to the nanoplanes.

Figure 3 indicates that the side-scattering direction is orthogonal with respect to the incident polarization. When the incident polarization is perpendicular to the nanoplane, the scattering happens mainly in the direction parallel to the nanoplane, with no component orthogonal to the plane, whereas when the incident polarization is parallel to the nanoplane, the light is mainly scattered in the direction perpendicular to the nanoplanes (however, with less efficiency). This suggests a strong anisotropy in the optical properties and directionality of light interaction.

We have also tested the spectral response of the scattering phenomenon. Figure 4 shows that the simulated scattering cross sections agree with the theory of Rayleigh scattering, in which the total scattering intensity is inversely proportional to the 4th order of the wavelength. Across the simulation wavelength window, the scattering for incident polarization perpendicular to the nanoplane is visibly stronger than that for incident polarization parallel to the nanoplane.

The scattering cross sections at 800 nm wavelength are 3.18253 (nm^2^) and 1.80742 (nm^2^) for perpendicular and parallel polarization, respectively. The superposition of scattering caused by each individual sheet contributes to the total propagation loss of type II waveguides, so that the mode with polarization perpendicular to the nanogratings suffers much more scattering loss than that with polarization parallel to the nanogratings. The directionality of scattering in the type II waveguide will be further confirmed by the following experimental results.

### 2.3. Scattering Experiments

In order to verify the polarization dependency of scattering and the directivity of scattering, an experiment was carried out by monitoring the optical power leaking out from the waveguide. In this experiment, a type-II waveguide consisting of a smooth positive index core and nanostructured cladding was generated [3]. Phase-contrast microscopy (PCM) images are presented in Figure 5a and Figure 6a. Via an imaging technique, the PD scattering phenomenon was observed for the type II waveguide upon 800 nm light injection (Figure 5b,c and Figure 6b,c). The scattering direction is dominantly orthogonal to the incident polarization. A rotation of the injection polarization state also results in a rotation of the scattering direction.

When vertically polarized light is injected into the type II waveguide written with vertical polarization (which produces horizontal nanoplanes), the scattering takes place with the maximum intensity in the horizontal direction. Otherwise, when horizontally polarized light is injected into the same type II waveguide, the scattering occurs primarily in the vertical direction. In the experiment, if the scattered light is detected in the vertical direction, the overall scattering intensity for horizontal polarization injection is therefore stronger than the scattering intensity for vertical polarization injection, as indicated in Figure 5d. The reason lies in the fact that although the total scattering for horizontal polarization injection along the nanoplane is weaker than that for vertical polarization injection (perpendicular to the nanoplanes), the scattering happens mainly in the vertical direction.

For type II waveguides written by horizontally polarized laser radiation (vertical nanoplanes), when the scattering is detected in the vertical direction, the scattering intensity for horizontal polarization injection is much stronger than the scattering intensity for vertical polarization injection, as shown in Figure 6. This is because total scattering for horizontal polarization injection is stronger than total scattering for vertical polarization injection, and scattering happens mainly in the vertical direction for horizontal polarization injection.

For the sake of clarity, a comparison of the scattering intensity measured in the vertical direction corresponding to the two orthogonal injection polarization is given in Table 1. Considering the relation between writing polarization and the orientation of nanoplanes, the measured PD scattering characteristics are consistent with the PD scattering behavior predicted by FDTD simulation.

## 3. Application

It has been reported that a type II waveguide has a strong polarization-dependent loss, and, specifically, has a lower propagation loss for optical signals with polarization parallel to the nanogratings. Therefore, it can be utilized as a waveguide polarizer. However, its minimum propagation loss for the preferred polarization state is typically as high as 6 dB/cm. As is mentioned in [11], the type II waveguide is a particular kind of type II traces, hosting a positive core index. Under certain writing conditions, the cross section can be filled completely by nanogratings, so that this kind of type II trace cannot guide light. It is different from the cases in chalcogenide glass and aluminosilicate glass [22,23]. The type II waveguide is called type II-WG and the non-guiding type II trace is called type II-NG.

As the PD scattering is an efficient way to create polarization-dependent loss, we present below a concept that allows preserving the typical low losses of a type I waveguide while gaining additional polarization sensitivity provided by type II traces outside the type I waveguide. It is straightforward to predict that if two non-guiding type II-NG traces are written around a type I waveguide with their mode areas overlapping to some extent, the evanescent field of a type I waveguide can be modified by the presence of the type II trace and its anisotropic scattering. Therefore, a low-loss waveguide polarizer can be realized by utilizing the polarization-dependent scattering loss of the type II trace in the vicinity of the type I waveguide.

The proposed concept and its realization of the low-loss waveguide polarizer are shown in Figure 7a–c. Two type II-NG traces are placed on both sides of a type I waveguide. The 8.7 mm-long type I waveguide was written by 800 nm laser pulses with 150 mW average power, 100 kHz repetition rate, and 250 fs FWHM pulse width. The optical power was focused by a 0.42NA objective at a scanning speed of 100 μm/s in a longitudinal writing scheme. The 4.35 mm-long type II-NG trace written on each side of the type I waveguide was fabricated using 250 mW average power, 100 kHz pulse repetition rate, and 350 fs pulse width. The laser pulses were vertically polarized and the scanning speed was 20 μm/s. The separation between the center of the type I and type II traces is 5 μm. The test of the composite optical waveguide properties was made by using a linearly polarized 800 nm optical signal as the probe. Figure 7b depicts the PCM image of the waveguide polarizer, and Figure 7d,e show the near-field mode when the probe optical signal was horizontally and vertically polarized, respectively.

Figure 7 reveals that the horizontally polarized optical signal can be guided with relatively low propagation loss while the vertically polarized optical signal is suppressed. Considering that in this case the nanograting planes are horizontally oriented (determined by the vertical polarization of the writing laser pulses), only the signal light, which is parallel with the nanograting planes, can be guided while the perpendicularly polarized signal light is damped. This is consistent with the simulation results previously discussed, which predicted a strong polarization sensitivity and a higher scattering rate for polarizations perpendicular to the nanoplane.

The measured polarization-dependent propagation loss and the subsequent polarization guiding is a direct consequence of the polarization-selective scattering by the nanogratings within the type II-NG structures in the vicinity of the type I waveguide. The interaction between type I and type II traces becomes more obvious with the decrease of the separation between them due to evanescent field modulation. Subsequently, in order to systemically measure the strength of evanescent mode field interaction and the impact of the nanogratings, a few waveguide polarizers were fabricated with separations between the type I and the type II traces ranging from 5 um to 12 um. Figure 8 shows the dependence of the extinction ratio on the separation between type I and type II traces. For the 8.7 mm long waveguide, the polarization extinction ratio can be as high as 20.9 dB with 5µm traces separation, and the loss is expected to be lower than 1 dB. This extinction ratio decreases exponentially with the increase of the trace separation. This is attributed to the exponential decay of the evanescent mode field away from the waveguide core, as indicated by a red line in Figure 8.

## 4. Conclusions

The mechanism of PD guiding in type II waveguides was investigated, indicating that guiding occurs only for the mode with polarization parallel to the nanograting planes. A scattering mechanism was proposed to illustrate the different transport efficiencies for the orthogonally polarized modes.

FDTD simulation of PD scattering demonstrated that scattering for an optical signal with field polarization perpendicular to the nanoplanes is stronger than that parallel to the nanoplanes, which is the primary PD loss mechanism in addition to the intrinsic birefringence. Based on the polarization-dependent scattering properties of nanogratings created by femtosecond laser pulses, we have successfully demonstrated a low-loss waveguide polarizer in fused silica with a >20 dB extinction ratio. This composite waveguide polarizer was realized by combining a type I waveguide with type II laser-written traces, and the polarization selectivity relies on the interaction between the evanescent field of the type I waveguide and polarization-dependent scattering of type II traces adjacent to it.

## Figures and Tables

**Figure 1 materials-15-05698-f001:**
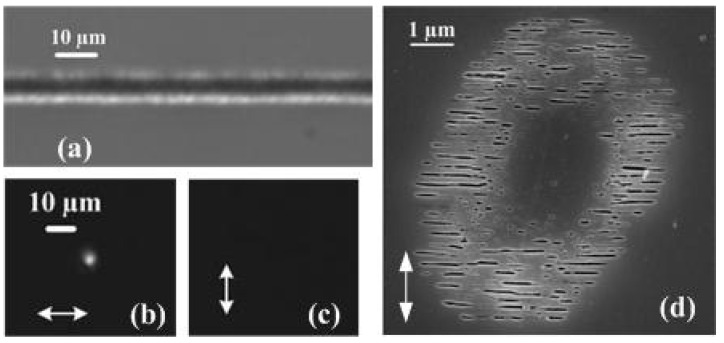
Polarization-dependent guiding of type II waveguides written by vertical polarization. (**a**) shows top-view positive phase contrast microscope (PCM) images of type II waveguides. The structure is written by 300 mW, 150 fs, 100 kHz laser pulses at the speed of 10 µm/s. The length of the waveguide is 7.3 mm. The writing and injection are all performed via an objective with 0.28NA. (**b**,**c**) show near-field modes of the waveguide (**a**) for injected 800 nm radiation. (**b**) shows the case where injection has horizontal polarization, and (**c**) is the case with vertical polarization. (**d**) SEM image of the cross-section of the waveguide.

**Figure 2 materials-15-05698-f002:**
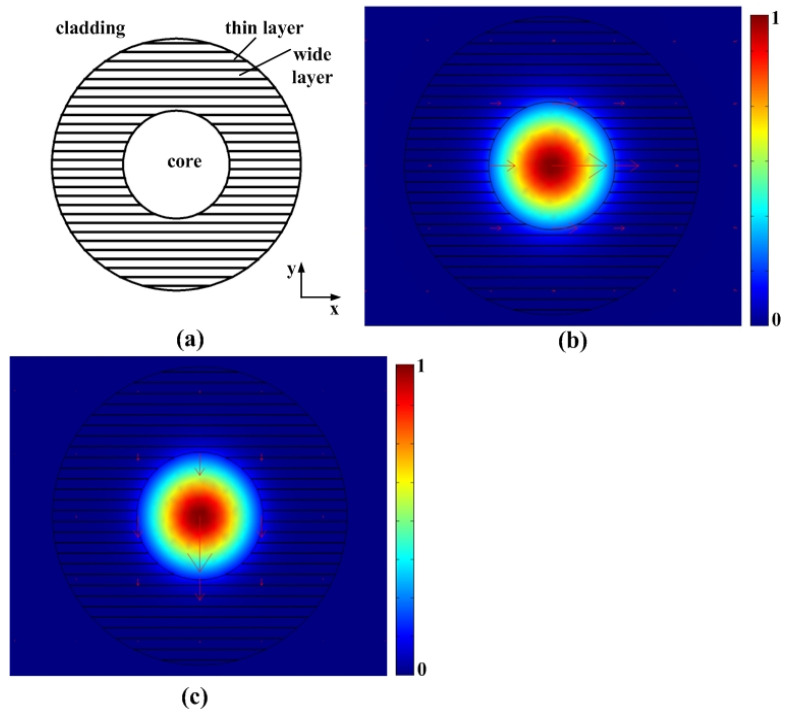
(**a**) The ideal model of the cross-sectional refractive index profile of a type II waveguide. (**b**) The z component of time average power flow profile for the x-polarized mode. (**c**) The z component of time average power flow profile for the y-polarized mode. The arrows in (**b**,**c**) indicate the direction of the transverse electrical field. The length of the arrows is proportional to the amplitude of the electrical field.

**Figure 3 materials-15-05698-f003:**
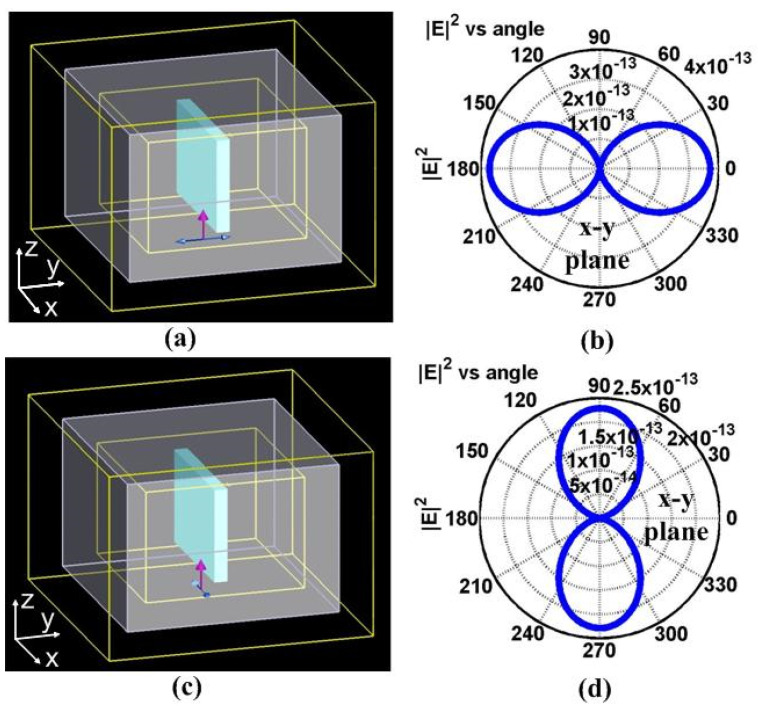
Schematic representation of the FDTD case model with the scattered intensity patterns in the far-field zone. The source polarization in (**a**,**c**) is perpendicular and parallel to the nanoplane, respectively. (**b**,**d**) show the scattered intensity in the far-field zone (1 mm away from the center) in the x-y plane, which crosses the center of the cuboid. The 0 degree corresponds to the x axis and the 90 degree corresponds to the y axis.

**Figure 4 materials-15-05698-f004:**
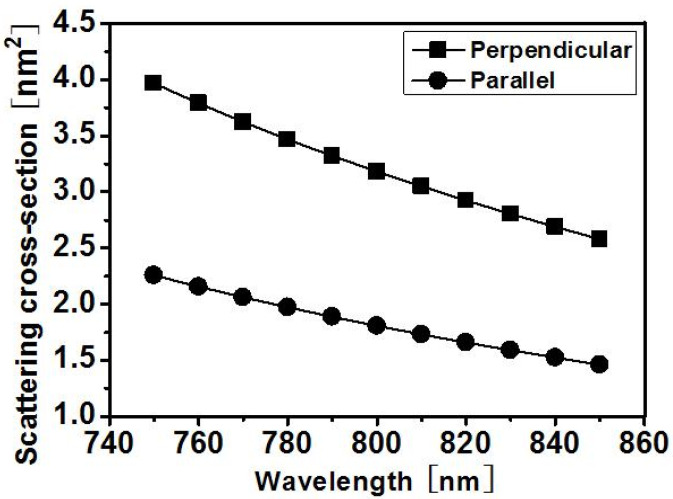
The total scattering cross section of an oblate cuboid calculated by the FDTD method as a function of wavelength. “Perpendicular” denotes polarization perpendicular to the nanoplane and “Parallel” denotes polarization parallel to the nanoplane.

**Figure 5 materials-15-05698-f005:**
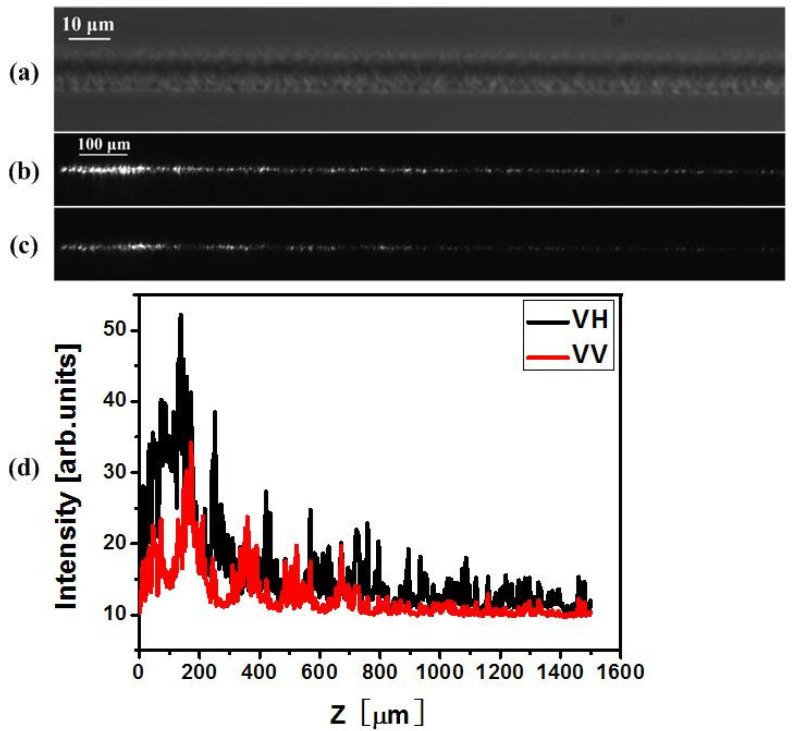
PD scattering intensity maps of type II waveguides detected in the vertical direction. The waveguide is written by 10 mW, vertically polarized, 1 kHz laser pulses at a scan velocity of 1 μm/s. The length of the waveguide is 1.5 mm. (**a**) top view PCM image of type II waveguide. (**b**,**c**) top view images of the scattering properties over the whole length. (**b**) in this case the trace is injected by a horizontally polarized field, whereas (**c**) is injected by vertically polarized light at 800 nm. (**d**) shows the vertical scattering yield as a function of the z position along the trace. The letters on the right indicate writing/injection polarization.

**Figure 6 materials-15-05698-f006:**
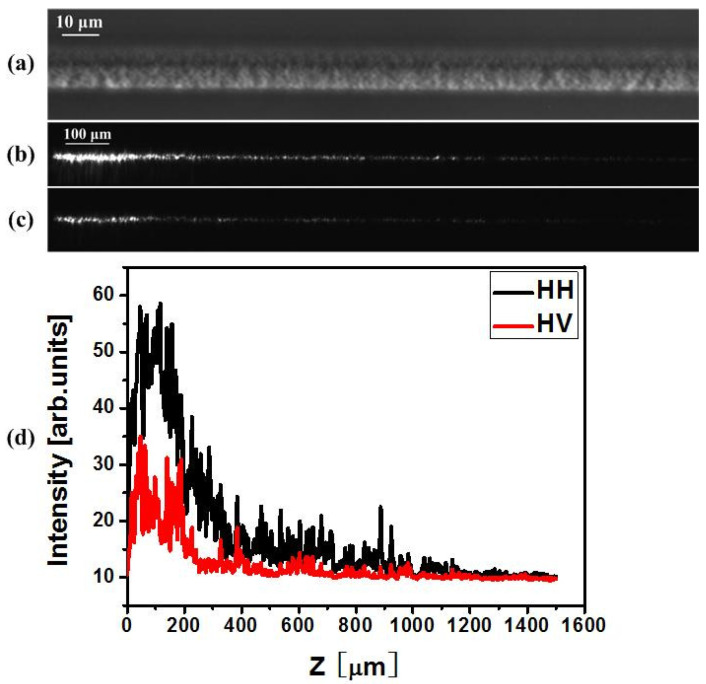
PD scattering intensity maps of type II waveguides detected in the vertical direction. The waveguide is written by 10 mW, horizontally polarized, 1 kHz laser pulses at a scan velocity of 1 μm/s. The length of the waveguide is 1.5 mm. (**a**) top view PCM image of type II waveguide. (**b**,**c**) top-view images of the scattering properties over the whole length. (**b**) in this case the trace is injected by a horizontally polarized field, whereas (**c**) is injected by vertically polarized radiation at 800 nm. (**d**) shows the scattering intensity as a function of the z position along the trace. H denotes horizontal direction and V denotes vertical direction. The letters on the right indicate writing/injection polarization.

**Figure 7 materials-15-05698-f007:**
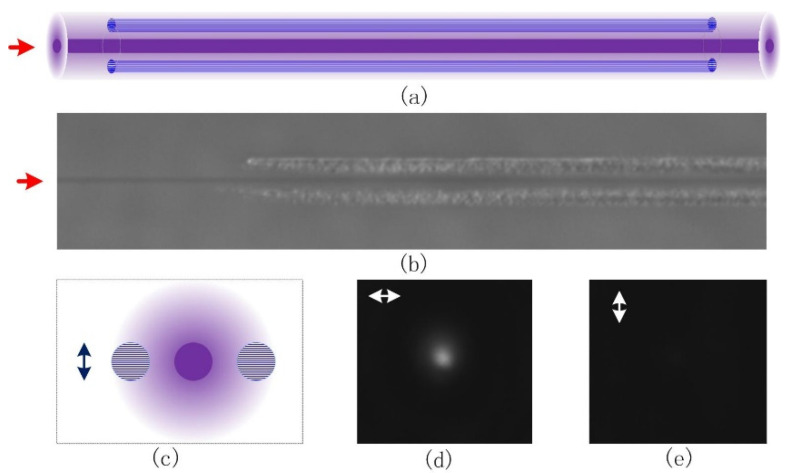
Polarization-dependent optical guiding in a waveguide polarizer in fused silica written by vertically polarized 800 nm femtosecond laser pulses. (**a**) Conceptual model of the proposed structure, (**b**) PCM side view of a real structure, the central type I waveguide is written with 150 mW, 100 kHz, 250 fs laser pulses at the speed of 100 μm/s, and type II is realized with 250 mW, 100 kHz, 350 fs laser pulses, and 20 μm/s writing speed. (**c**) Schematic end view of the structure; the arrow shows the writing polarization. (**d**,**e**) Near-field modes with horizontally and vertically polarized laser injection, respectively.

**Figure 8 materials-15-05698-f008:**
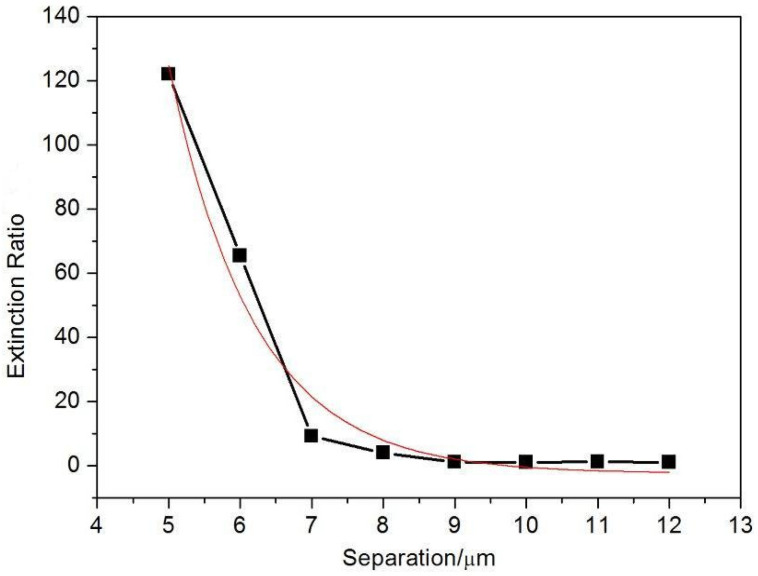
The dependence of the extinction ratio (ER) between orthogonal modes on the separation between type I and type II traces. The type I waveguide is 8.7 mm long, written with 150 mW, 250 fs laser pulses at the speed of 100 μm/s. The type II-NG is 4.35 mm long, written with 250 mW, 350 fs laser pulses at the speed of 20 μm/s. A quasi-exponential decay with increasing the separation distance is observed. ER = 10log(P_horizontal_/P_vertical_) here.

**Table 1 materials-15-05698-t001:** Comparison of the scattering intensity in the vertical direction between the orthogonal injection polarization. H denotes the horizontal direction and V denotes the vertical direction. For the horizontal nanogratings, IH denotes scattering intensity in the vertical direction for horizontal polarization injection, whereas IV denotes the vertical scattering yield for vertical polarization injection. This is similar for IH * and IV * in the case of vertical nanogratings.

Direction of Writing Polarization	Direction of Nanoplanes	Direction of Injection Polarization	Scattering in Vertical Direction	Main Scattering Direction
V	H	H	I_H_	V
		V	I_V_ (<I_H_)	H
H	V	H	I_H_ *	V
		V	I_V_ * (<I_H_ *)	H

## Data Availability

Not applicable.

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
