# Peer review of "Polarization-Dependent Scattering of Nanogratings in Femtosecond Laser Photowritten Waveguides in Fused Silica"

_materials, 2022, doi:10.3390/ma15165698_

Round 1
Reviewer 1 Report
In this article, the authors showed the characteristics structural and optical properties of laser photowritten waveguides in fused silica. The finding shows that the polarization selectivity in the optical fiber comes from the microscopic structural origins, which can be controlled by laser polarization. The results are interesting and useful, but the manuscript needs some improvements as detailed below:
1. The authors should clarify the definition of “type II waveguide” in the manuscript, as this is not a common and well-accepted terminology. It seems this definition comes from earlier publications from the same authors, but it is suggested that this article can be understood by the audience without the necessity to check previous papers.
2. There are quite some errors in the symbols of the manuscript. For example, strange symbols are displayed on lines 99, 102, 145, and 229. Careful proofreading of the manuscript is necessary.
3. The definition of the “extinction ratio” in figure 8 is not given, which should have been included.
4. For Figures 5 and 6, the meaning of “HH” and “HV” should be clearly explained in the figure legends.
5. The section number is not labeled correctly. The introduction is labeled as section 0, which is not correct based on the later subsection numbers.
Author Response
Comment 1
- The authors should clarify the definition of “type II waveguide” in the manuscript, as this is not a common and well-accepted terminology. It seems this definition comes from earlier publications from the same authors, but it is suggested that this article can be understood by the audience without the necessity to check previous papers.
Our Reply to Comment 1:
Thank the reviewer for the valuable comments. We rewrote the first paragraph partly as following:
Typically, the modifications in fused silica can be divided into two categories: type I waveguide and type II waveguide. The type I waveguide is written at low pulse radiation levels, the modification is quasi-isotropic, leading to a smooth refractive index increase of about 10-3, its losses typically lower than 0.2 dB/cm, which can be useful for photonic device fabrication. The type II waveguide is written at higher pulse energy levels, the laser-induced modification may contain an arrangement of periodic nanoplanes which align themselves orthogonally to the laser polarization, which can be utilized to form type II waveguides [3] with polarization dependent guiding properties.
Comment 2
- There are quite some errors in the symbols of the manuscript. For example, strange symbols are displayed on lines 99, 102, 145, and 229. Careful proofreading of the manuscript is necessary.
Our Reply to Comment 2:
We have modified corresponding part in the manuscript and marked them in blue.
We deleted the @= on line of 99, and replaced the wrong symbols by 1.5 mm and 3.5 mm on line of 102, by dscat(w)=Pscat(w)/Isource(w) on line of 145, and by 1mm/s on line of 229.
Comment 3
- The definition of the “extinction ratio” in figure 8 is not given, which should have been included.
Our Reply to Comment 3:
Thank the reviewer for the valuable comments.
We put on “ER=10log(Phorizontal/ Pvertical) here” on line of 306.
Comment 4
- For Figures 5 and 6, the meaning of “HH” and “HV” should be clearly explained in the figure legends.
Our Reply to Comment 4:
Thank the reviewer for the valuable comments.
We put on “H denotes horizontal direction and V denotes vertical direction” on line of 232.
Comment 5
- The section number is not labeled correctly. The introduction is labeled as section 0, which is not correct based on the later subsection numbers.
Our Reply to Comment 5:
Thank the reviewer for the reminder. We corrected the section number, sorting from No.1.

Reviewer 2 Report
In this paper, the authors present a systemic investigation on the fundamental mechanism, the performance, and the applicability of laser-triggered nanogratings. In order to explain the underlining mechanism of PD scattering, finite-difference time-domain (FDTD) simulation was conducted. The simulation results confirm that the scattering cross-section of the nanopatterned trace is highly sensitive to the injected polarization. PD scattering was further analyzed in experiments, indicating that scattering is the major source of polarization-sensitivity behind the observed wave guiding behavior of type II waveguides, while form birefringence is a relatively minor contributor. Coupling mixed regimes of type I and type II, a low loss waveguide polarizer is demonstrated based on the modulation of the evanescent field near type I waveguide through polarization dependent scattering of nanogratings.
The paper is interesting, well written and well structured; the text is clear and easy to read, and the results are sufficiently discussed. Overall, the manuscript is well thought out and written, the objectives clearly stated, simulation and experimental methods are advanced, data statistically analyzed, the conclusions well supported by the data presented. This makes the paper possible for publishing in the Materials. However, to accept, the paper should be subject to some revision. The detailed remarks are as follows:
1. In the introduction section, please provide more researches that have been done in this field in order to provide sufficient background and relevant references. Sufficient information about the previous study findings should be added and compared with the present study so that the readers could follow the present study rationale and procedures.
Author Response
Thank you very much for positive comments and valuable comments. Following the suggestion, we put on some new results on nanograting generation and some applications with corresponding references.
Beside the form birefringence and polarization-sensitive scattering, diffraction of nanograting has been exploring recently. Diffraction gratings at several centimeters on fused silica and silicon wafer with Ag film have been reported [13,14]. Most cases nanogratings induced by fs laser pulses have been observed in fused silica and some derivates, rarely in other optical glasses. Considering a potential choice for fabricating polarizing elements in integrated photonic circuits, other optical materials such as Ti:sapphire[15], Yttrium Aluminum Garnet (YAG)[16], and Diamond-like carbon[17] enable the formation the nanograting irradiated by polarized ultrashort pulse laser.
And some refence:
- CHEN L, CAO K, LI L, et al. Large-area straight, regular periodic surface structures produced on fused silica by the interference of two femtosecond laser beams through cylindrical lens[J]. Opto-Electron Adv,2021, 4(12):200036. DIO:10.29026/oea.2021.200036
- GENG, J., YAN, W., SHI, L. et al. Surface plasmons interference nanogratings: wafer-scale laser direct structuring in seconds[J]. Light Sci Appl., 2022, 11, 189 (2022). DIO:10.1038/s41377-022-00883-9
- BAI J, CHENG G, LONG X, et al. Polarization behavior of femtosecond laser written optical waveguides in Ti:Sapphire[J]. Optics Express, 2012, 20, 15035-15044. DIO: 10.1364/OL.30.001285
- REN Y, ZHANG L, ROMERO C, et al. Femtosecond laser irradiation on Nd:YAG crystal: Surface ablation and high-spatial-frequency nanograting[J]. Applied Surface Science, 2018, 441: 372-380. DIO: 10.1016/j.apsusc.2018.01.217.
- POPESCU C, DORCIOMAN G, BITA B, et al. Fabrication of periodical surface structures by picosecond laser irradiation of carbon thin films: transformation of amorphous carbon in nanographite[J]. Applied Surface Science, 2016, 390: 236-243. DIO:10.1016/j.apsusc.2016.08.029.
